# The General Public and Young Adults’ Knowledge and Perception of Palliative Care: A Systematic Review

**DOI:** 10.3390/healthcare12100957

**Published:** 2024-05-07

**Authors:** Yann-Nicolas Batzler, Manuela Schallenburger, Jacqueline Schwartz, Chantal Marazia, Martin Neukirchen

**Affiliations:** 1Interdisciplinary Centre for Palliative Medicine, Medical Faculty and University Hospital Düsseldorf, Heinrich-Heine-University Düsseldorf, 40225 Düsseldorf, Germany; yann-nicolas.batzler@med.uni-duesseldorf.de (Y.-N.B.); jacqueline.schwartz@med.uni-duesseldorf.de (J.S.); martin.neukirchen@med.uni-duesseldorf.de (M.N.); 2Department of the History, Philosophy and Ethics of Medicine, Centre for Health and Society, Medical Faculty, Heinrich-Heine-University Düsseldorf, 40225 Düsseldorf, Germany; chantal.marazia@med.uni-duesseldorf.de; 3Department of Anesthesiology, Medical Faculty and University Hospital Düsseldorf, Heinrich-Heine-University Düsseldorf, 40225 Düsseldorf, Germany

**Keywords:** knowledge, palliative care, perception, public health, stigma, young adults

## Abstract

*Background:* As a result of demographic change, chronic and oncological diseases are gaining importance in the context of public health. Palliative care plays a crucial role in maintaining the quality of life of those affected. International guidelines demand access to palliative care not only for the elderly but also for younger people who face severe illnesses. It can be assumed that palliative care will become increasingly important for them. In order to develop public health strategies which are able to promote palliative care, it is important to assess the knowledge of, and attitude towards, palliative care as found among members of the general public and its specific target groups. In particular, little is known about young adults’ knowledge and perceptions of palliative care. *Objectives and design:* This work aimed to assess the understanding and viewpoints regarding palliative care among the general population and among young adults aged 18 to 24. We therefore conducted a systematic review, which, for this target population, could be seen as a novel approach. *Methods:* Exclusion and inclusion criteria were developed using the PICOS process. Literature was researched within MEDLINE (via PubMed), Google Scholar and Web of Science. A search string was developed and refined for all three databases. Grey literature was included. Duplicates were excluded using Mendeley. The literature was independently screened by two researchers. Narrative synthesis was used to answer the main research question. *Results:* For the general public, palliative care is still associated with death and dying and comforting sick people towards the end of their lives. Multiple social determinants are linked to better knowledge of palliative care: higher education, higher income, female gender, having relatives that received palliative care, and permanent employment. The population’s knowledge of palliative care structures increases, the longer such structures have been established within a country. Young adults are familiar with the term palliative care, yet their understanding lacks nuance. They associate palliative care with death and dying and perceive palliative care to be a medical discipline primarily for the elderly. Nevertheless, young adults demand participation within the planning of interventions to destigmatize palliative care. *Conclusions:* The general public still lacks a detailed understanding of palliative care. Palliative care faces stigma at multiple levels, which creates barriers for those who set out to implement it. However, addressing young adults as a crucial peer group can help break down barriers and promote access to palliative care.

## 1. Introduction

As a result of demographic change, chronic and oncological diseases are gaining importance in the context of public health [1]. These disease entities often lead to a multitude of severe symptoms affecting the quality of life of those affected. Palliative care plays a crucial role in alleviating this kind of substantial symptom burden. The World Health Organization describes palliative care as an “approach that improves the quality of life of patients (adults and children) and their families who are facing problems associated with life-threatening illness” [2]. However, palliative care is often implemented with restraint [1]. This has a variety of reasons.

For centuries, death and dying were perceived as being part of daily life [3,4,5]. As the process of dying became increasingly medicalized, death was removed from its common place in home settings and was shifted towards institutional environments such as hospitals and hospices [4]. Hospices date back to early Christianity and originally served as pilgrimage sites [6,7,8]. With the process of industrialisation, the overall age of people increased and chronic diseases gained importance [7,8,9]. Due to medical advances, oncological diseases have the potential to be treated curatively. In the setting of incurable diseases, thorough research has led to improvements in symptom control and death does not come as unexpectedly as in earlier times [9]. Palliative care originated alongside these developments in the 1960’s in England and Canada. With the concept of “Total Pain”, Cicely Saunders, one of the pioneers of palliative and hospice care, established that suffering is not only experienced on a physical level, but also on a psychological, social and spiritual level [9]. This corresponds to the basic idea of today’s palliative care. Be that is it may, the medical discipline of palliative care still faces stigmatization [10].

Link and Phelan’s Model of Stigmatization helps to explain this phenomenon [11,12]. In modern societies, death and dying have been pushed far out of everyday life. People who are seriously ill deviate from the norm of striving for health and are downgraded to illness (“Labelling”). In the past, care for these individuals was provided by hospices, and, more recently, by palliative care services. As a result, it is a common perception that palliative care exclusively focuses on the needs of seriously ill patients who are approaching the end of their lives (“Stereotyping”). Crucially, just as death and dying have been excluded from everyday life, palliative care has been excluded from societal life. This leads to structural discrimination, which in turn means that patients who would benefit from an integration of palliative care do not make use of it due to fear of exclusion from social life [11,13,14,15].

In the context of public health, stigmatization affects health behaviour: health services are used less frequently and treatments are delayed. Stigmatized patients also show poorer compliance and adherence to treatment [16,17]. Medical staff are also unable to completely distance themselves from the stigmatizing processes, which can lead to poorer quality of treatment. This increases morbidity and mortality, which further promotes social inequality [16,17,18]. Gaining insight into the public’s knowledge and perception of palliative care is crucial for reducing the stigma surrounding palliative care and promoting its integration into regular clinical practice [16,17,18,19,20,21].

This study aimed to assess the current knowledge and perception of palliative care among the general population and among young adults aged 18 to 24 years. To this purpose, we conducted a systematic review of the existing literature.

## 2. Methods

This systematic review was conducted according to the “Preferred Reporting Items for Systematic Reviews and Meta-Analyses” (PRISMA) statement [22]. It is registered on protocols.io: dx.doi.org/10.17504/protocols.io.261gedymwv47/v2 (accessed on 18 March 2024).

Before determining the search strategy, the central research question was defined and concretized using the PICOS process. To include a multitude of studies, the general population was included, with young adults serving as a subpopulation. The German Federal Ministry of Justice and Consumer Protection defines 18- to 27-year-olds as “young people” [23]. There is no internationally agreed definition; therefore, this age range was adapted for this study: people aged 18 to 24 were defined as the target population of “young adults”. They are an ideal target population in the context of public health, given that their health behaviour can still be influenced. In addition, health-related topics are relevant for them via discussions with family members and peers [24,25]. Focusing on this target group’s perception and knowledge of palliative care in a systematic review is a novel approach that potentially helps in promoting palliative care.

Table 1 summarizes the target population, interventions, primary endpoints (outcomes) and types of included studies/publications (study design), and exclusion criteria.

The databases searched for the systematic review were MEDLINE (via PubMed), Google Scholar and Web of Science. A specific search strategy was developed for each database. Google Scholar was included in the search to capture grey literature. Since our study group was primarily German speaking, a German search string was developed alongside an English one for Google Scholar. The search strings for all three databases can be found in the Appendix A. With the help of Mendeley (Mendeley Ltd., New York, NY, USA, Version 2.99.0, 2023) the publications matching the search strings were checked for duplicates and the literature was managed. A sensitive research method was used for this systematic review.

Suitable literature that met the inclusion and exclusion criteria was screened by two researchers (Y.B. and M.S.). After a joint decision on which literature to eventually incorporate, each article was independently screened by the two researchers. The main findings were recorded in a table. Whenever main results were not in accordance, for example, due to unclear study design or statistical analysis, a third researcher (M.N.) served as a consultant. In that case, uncertainties were discussed. After completion and adaption of the main findings for each study, the findings were categorized with the help of a table of results (see Appendix A). On this basis, narrative synthesis was used to report on the main results that answered the research question.

## 3. Results

After searching the three databases, a total of 1766 articles were identified, whereupon a total of 435 duplicates were detected. One publication was excluded because it was only available in Mandarin. Each article was assessed for content based on title and abstract. A total of 1261 articles were excluded as they did not meet the inclusion criteria.

Two articles could not be retrieved in full versions. After reviewing the 67 remaining articles, a further 7 articles were excluded, given that healthcare professionals were the target population, or the primary outcomes “knowledge” and “perception of palliative care” were not analyzed. All results were unanimous among the researchers (see Appendix A). In the end, 60 articles were included in the systematic review. The PRISMA flowchart summarizes the process (see Figure 1).

### 3.1. Study Types, Years and Countries

Cross-sectional studies made up the largest portion (*n* = 39, 65.0%) of the articles. The literature reviews included one systematic review, two scoping reviews and one integrative review. Furthermore, one congress paper, one dissertation and one master’s thesis were included. No randomized controlled trials were found to answer the research question. Figure 2 summarizes the included types of studies.

The publications ranged from the years 2008 to 2023. Most of the articles were published between 2020 and 2023, followed by publication dates between 2016 and 2019. Their distribution is shown in Figure 3. Studies from different continents were included in the review, and their distribution is shown in Figure 4.

On the basis of the main findings for the general population, a narrative synthesis of young adults’ knowledge and perception of palliative care answered the main research question. For both groups, the main aspects focused on:Assessment of knowledge;Knowledge and establishment of palliative care;Knowledge and sociocultural determinants;Perceptions and associations;Resources to obtain information on palliative care.

### 3.2. General Population: Knowledge and Perception of Palliative Care

The main results of the systematic review are graphically summarized in Figure 5.

Knowledge of palliative care was assessed using several different questionnaires. Among others, the validated PaCKS questionnaire was used. This numerically depicts knowledge of palliative care on the basis of 13 items. Participants need to decide if each stated item is true or false with the option of answering “I don’t know” (then coded as a wrong answer). The sum score adds up to a maximum of 13 (= highest knowledge) [26,27].

In a study from the United States, 301 participants were included. Their mean PaCKS score was 5.25 and many answered with “I don’t know” [28]. A comparable result was found in another study (*n* = 154) from the United States, in which a mean of 4.38 was reached [29]. Another American study found a mean score of 9.9 prior to an online 60 min educational course. This course led to a significant rise in knowledge (PaCKs post: 12.8) [30]. The lowest PaCKs scores were reached in a study from Jordan (*n* = 430, PaCKS mean: 3.51) [31].

Furthermore, it was recorded whether respondents had already heard of palliative care, had no previous knowledge of palliative care or had precise knowledge. In a study from Italy (*n* = 1897), 59.4% of respondents had heard of palliative care, but only 23.5% had detailed knowledge [32]. In studies from Northern Ireland, most respondents had heard of palliative care (83%), but only a small number of them had thorough knowledge (20.1%) [33,34,35]. In a study from the Netherlands (*n* = 1242), 90% had heard of palliative care, but only 47% stated that they had precise knowledge. When specific knowledge was assessed within this study, only 29.8% of respondents knew that palliative care means no exclusion of other therapies such as radiation. Furthermore, 23.5% of respondents knew that palliative care is not only for patients in the last few weeks of their lives [36]. In a study from Sweden, 41% of respondents had not known what palliative care was; 43% had heard of palliative care, however, they only had superficial knowledge [37].

In summary, there is a lack of nuanced knowledge within the general population [32,33,38,39,40,41]. Participants were not able to specifically name the services included in palliative care, such as the care of relatives or the approach to symptom control on multiple, for example, spiritual or psychological, dimensions [42,43,44].

The countries of origin of the included studies were heterogeneously distributed. In countries like Jordan, where palliative care structures had not been established for long, there was less knowledge of palliative care. A direct comparison of two studies revealed the greatest discrepancy in terms of knowledge of palliative care: 90% of Dutch respondents had heard of palliative care, whereas 78.6% of respondents in Jordan had never heard of it [31,36].

Throughout the included studies, it was shown that ethnic minorities, in particular, such as African Americans or Hispanics, had a lower level of knowledge than the comparison group, for example, the Caucasian populations [45]. Depicting the results of two included studies, 74% of African Americans and up to 84.5% of Hispanic Americans had no knowledge of palliative care [46,47]. Furthermore, certain social determinants were found to influence the knowledge of palliative care, as found among respondents. These included a higher level of education, for example, university or college degrees; a higher monthly net income, and permanent employment. If relatives or friends of respondents had previously been in contact with palliative care services, respondents showed greater amounts of knowledge of palliative care services, and its many ways of providing help in severe situations [31,33,36,38,47,48,49,50,51,52,53,54]. Other factors that influenced knowledge of palliative care were female gender and older age. It was primarily the age group of people aged 45 to 64 that showed more knowledge compared with younger age groups [31,37,46,47,55,56,57,58,59,60,61].

The perception of palliative care, like knowledge of palliative care, differed greatly across national borders. The longer palliative care structures had been in place in a country, the higher the level of awareness of palliative care among the general population [62,63,64,65,66]. This was reflected in the general interest in, and discussion of, palliative care as confirmed by an analysis of search queries via Google—in countries in which palliative care structures had been established and enshrined in the law, the volume of search queries on palliative care was higher than in countries that were lagging behind in this regard [62].

Regarding palliative care, there is still an association with pure end-of-life care and with hospice care [56,67,68,69,70,71,72,73]. The general population believes that the main aspect of palliative care is analgesia for severe pain, which is seen as particularly important for cancer patients and the elderly [32,37,39,43,57]. The general public perceives palliative care as caring for people who have no wish for treatment and no hope for continued life [57]. Thus, palliative care is associated with reduced medical options and inadequate care [74]. This perception of palliative care is linked to palliative care wards in general. Crucially, such wards are perceived as places where people go to die, and people are unsure as to whether their autonomy and dignity would be maintained on such palliative care wards [75]. Furthermore, it is often assumed that palliative care services are only available in such inpatient wards. Most people are not aware of the possibility of receiving care at home by specialised outpatient palliative care teams [75,76,77].

Cultural determinants also influence the perception of palliative care. In an Asian subpopulation in Canada, palliative care was found to be inconsistent with prevailing beliefs and cultural values for up to 44% of respondents, due to its association with the end of life [78]. In these groups, death, dying and the end of life are considered as particularly taboo and palliative care options are ignored [78,79].

### 3.3. Young Adults: Knowledge and Perception of Palliative Care

Three studies were identified whose specific target population was young adults [24,80,81]. Other studies only partially included data from 18- to 24-year-olds as secondary endpoints in their evaluations.

Although many young adults have heard of palliative care, the prevalence rates vary widely. In a study from the UK conducted among students enrolled in different faculties (50% from the faculty of Life and Health Sciences), up to 83% of respondents had heard of palliative care [81], compared with only 40.3% of young adults in a study from Bangladesh [80]. Among young adults, there were repeated gaps in knowledge about the content and aims of palliative care. For example, when the PaCKS questionnaire was used, item number 5 (“Palliative care is only for people with a life expectancy of less than six months”), in particular, was answered incorrectly. In the age range of 18 to 24 years, one study showed an average PaCKS score of 3.2 [28], while another study showed an average score of 8.3 [81]. It should be noted that this higher value was achieved in a population of college students aiming at higher educational degrees. However, this discrepancy shows a tendency, namely that young adults aspiring to higher levels of education have more knowledge of palliative care than a broader sample with diversified educational backgrounds, for example, young adults in Bangladesh. The study situation was also heterogeneous when genders were compared in terms of their knowledge. In the above-mentioned student population, female respondents tended to know more about palliative care than their male counterparts [81].

More than half of the respondents who were able to develop a concept of palliative care for themselves felt that this concept was not clear and should be refined [80]. They were not aware of the holistic approach of palliative care and, compared with older populations, were more likely to think that palliative care does not address the care of relatives and friends [57]. Knowledge of palliative care in this age group is strongly influenced by whether relatives have already had contact with palliative care structures [24,50].

Besides personal contact with palliative care structures influencing knowledge and perceptions, most respondents in this age group received information on palliative care via the Internet; only 14% received it from books and only 8% from healthcare workers [80]. Mass media makes up 12% of the information sources, and information is increasingly obtained through social media [24,57,80].

As in the general population, in the young adult age group palliative care is perceived as mainly dealing with death and dying. Due to a perceived hopelessness in this, palliative care is associated with negative emotions such as giving up [24,56]. Young adults assume that palliative care is only available to certain people with certain disease entities—in the above-mentioned study from Bangladesh, 78% of those surveyed said that palliative care was only for people suffering from cancer [80].

## 4. Discussion

Due to demographic change and the associated increase in the prevalence of chronic diseases as well as the development of new tumour-specific therapies that prolong the survival of cancer patients, palliative care is becoming increasingly relevant. While medical societies and specific guidelines repeatedly call for the timely integration of palliative care in the course of a disease, across many sections of the population, the perception and lack of knowledge of palliative care can be seen as a barrier to its use.

### 4.1. Barriers to Utilization of Palliative Care

In the past, sick people were cared for by relatives at home until the end of their lives, but, with the development of modern care, death and dying have increasingly shifted to healthcare institutions [7]. In the general population, palliative care is often considered synonymously with hospice care and therefore equated with end-of-life care [34,42,69]. Due to the taboo surrounding death and dying in modern societies, there is a prevailing fear that illnesses may lead to social exclusion through processes of stereotyping. Fear and misconceptions about palliative care contribute to the reluctance of those affected to make use of palliative care services [82]. These barriers were also identified by another systematic review [83].

This systematic review found that older people, in particular, have more knowledge about palliative care, but that this knowledge is superficial and lacks detail [57,58,60,61]. Another systematic review found similar results in terms of a lack of nuanced knowledge [84]. People with a higher level of education and a higher annual salary are better informed about palliative care [46,56,59]. In contrast, ethnic minorities were found to have less knowledge of palliative care than comparison groups [46,47]. Personal experiences, such as caring for sick relatives, go hand-in-hand with a better knowledge of palliative care [55]. Similar to the perception of palliative care in the general population, young adults believe that palliative care is only appropriate for terminally ill people in the last six months of their lives [24,80,81]. In this population, too, palliative care is associated with death and dying, and with hopelessness, which fuels fear of possible utilization [80].

### 4.2. Promoting Palliative Care

As shown in this systematic review, socio-cultural, socio-economic and ethnic factors play an important role within the processes of stigmatization of palliative care [46,56,78]. When aiming to promote palliative care services, the importance of respecting cultural determinants was highlighted by an interpretive synthesis [85]. An emphasis should be put on respecting patients’ religious beliefs. In this context, possibilities of palliative care, such as receiving strong pain medication, might not be wished for by patients. Furthermore, it should be noted that religion is a crucial factor in making decisions at the end of life—sedation or withdrawal of food and liquids might not be tolerable [85].

Therefore, campaigns should keep several aspects in mind: language barriers; specific roles of family members within social structures; perception of death and dying; a culture-sensitive definition of a “good death”; and access to healthcare services, especially for ethnic minorities. Tailoring interventions towards these specific needs and requirements is crucial.

In the context of public health, nuanced knowledge can serve as a gateway to a more frequent use of palliative care services [83]. However, it seems unrealistic to suppose that an intervention to destigmatize palliative care will reach all strata of the population at the same time. Consequently, in public health campaigns, the focus currently has to be on clearly defined target groups.

In particular, young adults are still susceptible to a change in health behaviour through targeted interventions [24,25,81]. Since health behaviour can be altered in this group, interventions to destigmatize palliative care should focus on prevention and health promotion [86,87,88]. This includes a self-determined life and an early, forward-looking approach to end-of-life decisions. Drawing up advance directives while people are still healthy (living wills, health care proxies, or emergency ID cards, which record decisions in a compact format) creates a framework for deciding on desirable measures in the event of illness. Medical measures that do not correspond to people’s wishes could thus be prevented and medical and economic resources could be saved in the context of public health [89]. In addition to a forward-looking approach to end-of-life measures, low-threshold access to oncology services, knowledge of symptom control options and care structures in the event of sickness are essential in order to maintain quality of life and the ability to act. Therefore, the European Society for Medical Oncology (ESMO) and the European Society for Paediatric Oncology (SIOPE) work hand in hand in order to raise awareness of oncological diseases among young adults and improve the overall medical care for people in this age group [90]. Accordingly, awareness of the content of palliative care is crucial, especially in the setting of incurable diseases: if young adults are aware of the wide range of palliative care services (including outpatient clinics and at-home palliative care services), they can make use of them independently, even if primary care providers do not inform them in good time.

With the help of theory- and evidence-based health promotion models such as “Intervention Mapping”, interventional programs targeting young adults could be conducted in the near future. For example, interventions in primary or secondary educational institutions could have the potential to reach many students. Such interventions should make use of evidence-based methods of behavioural change. Through “interpersonal contact”, to name one example, knowledge could be disseminated through interaction with palliative care teams. The internet, especially social media platforms, should provide profound and easily accessible information on palliative care.

This empowerment should be an overarching goal of behavioural change campaigns in the context of public health in the near future [82].

### 4.3. Limitations

The methodology of this systematic review provided a suitable basis for researching international perceptions and knowledge of palliative care. Included studies were mainly cross-sectional, and thus were only able to provide a snapshot of knowledge and perceptions throughout different cultural areas. Only a few quasi-experimental studies in a pre-post design were found; these also only examined intervention effects over a short period of time. Although all results showed that young adults are not as well informed about palliative care as older population groups, this was generally not the specific main research question that was focused on in the literature.

The search string was developed in English and German (for Google scholar). Primarily English-speaking literature was used for this systematic review, since, in general, literature from different regions was available in English. To better screen for grey literature and country-specific research, however, search strings should be developed for more languages, such as Spanish or French.

This systematic review might be subject to publication bias methodology—it is possible that studies with positive results, especially when describing intervention results, are more likely to have been published than those with no positive findings.

The analysed publications were of heterogeneous regional distribution. This might contribute to bias in our findings since literacy rates might differ substantially. Furthermore, cultural determinants might lead to different health-aware lifestyles leading to bias when comparing results of the included studies. In summary, different healthcare systems might follow different approaches in establishing palliative care structures which makes comparisons difficult.

It was the aim of this systematic review to incorporate as many suitable studies as possible to answer the main research question. However, not many were found and study designs, numbers of participants, study lengths and primary endpoints varied. Therefore, a formal quality report was omitted in favour of a broad review of existing literature.

Studies with a control group and randomization, which are difficult to design due to the research question and target population, were not found. To fill this gap, longitudinal studies could help identify changes in attitudes and knowledge over time. Interventions with several components such as lectures, media presence, brochures and online campaigns were rarely carried out. A broad comparison of different interventions is difficult to achieve; however, this should be a key aim for future research.

## 5. Conclusions

Even after more than 60 years of continuous development of the palliative care sector, this young medical discipline still carries multiple stigma. The general population associates palliative care with death and dying as well as with medical care exclusively targeted to the end of life, which is perceived to be primarily available to oncology patients. Palliative care is still equated with hospice care. More specifically, young adults have often heard of palliative care, but they have no detailed knowledge of what palliative care really amounts to. At the same time, there are insufficient data on young adults’ awareness and knowledge of palliative care. What is certain, though, is that young adults receive little attention in interventions and public campaigns regarding palliative care, as offered in the context of public health. Such campaigns, however, would best be tailored towards clearly defined target groups—such as young adults—in order to promote and destigmatize palliative care. Young adults seem to be a suitable target population given that their health behaviour is still susceptible to change. Socio-cultural and ethnic factors should be recognised and information adapted accordingly. Future educational programs should pay attention to these factors and to the healthcare structures imposed by different healthcare systems, to unequal access to healthcare, to differences in literacy rates, and to different educational systems. This may lead to a broader acceptance of palliative care in all population strata in the near future.

## Figures and Tables

**Figure 1 healthcare-12-00957-f001:**
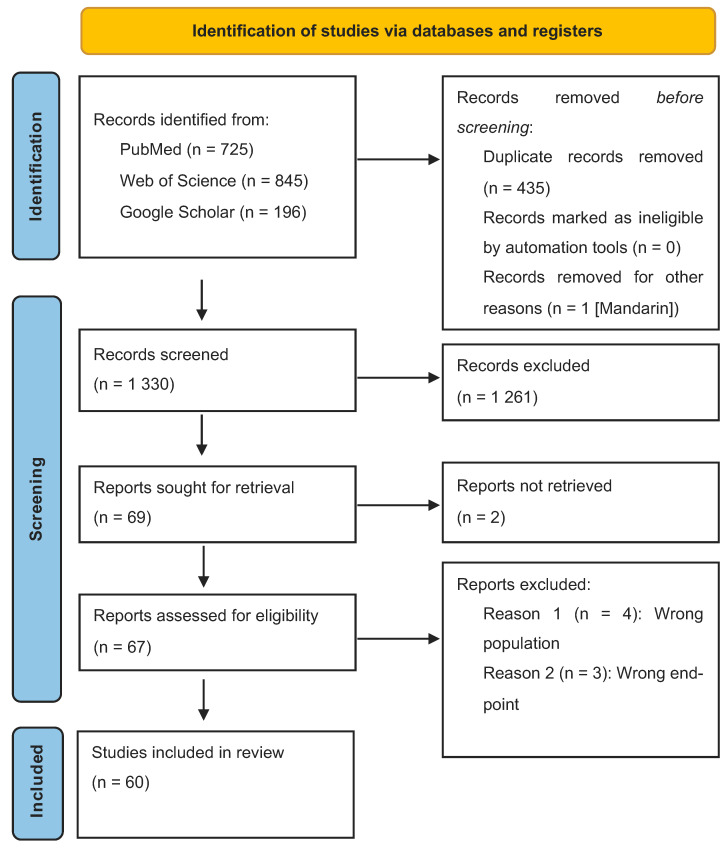
PRSIMA flow chart.

**Figure 2 healthcare-12-00957-f002:**
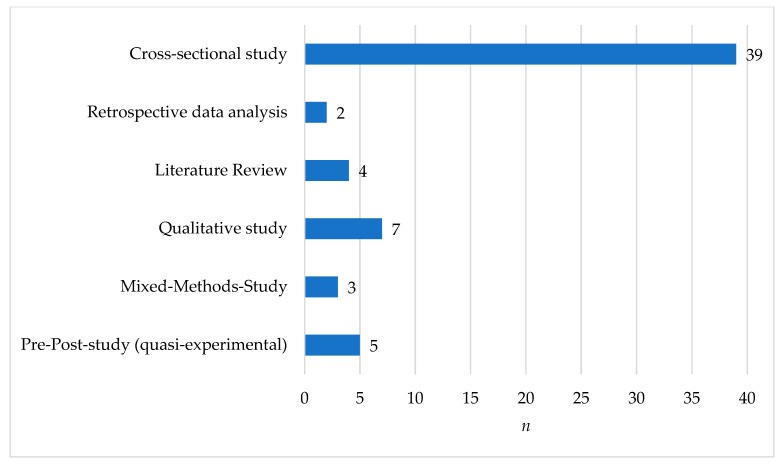
Distribution of studies used for the systematic review (*n* = 60).

**Figure 3 healthcare-12-00957-f003:**
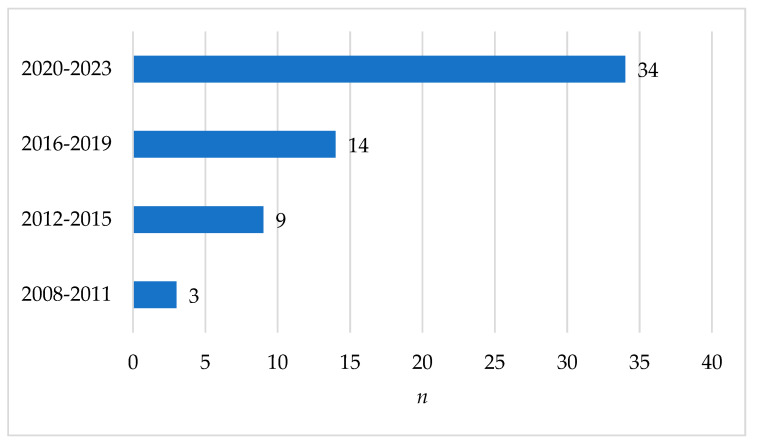
Articles and year of publication (*n* = 60).

**Figure 4 healthcare-12-00957-f004:**
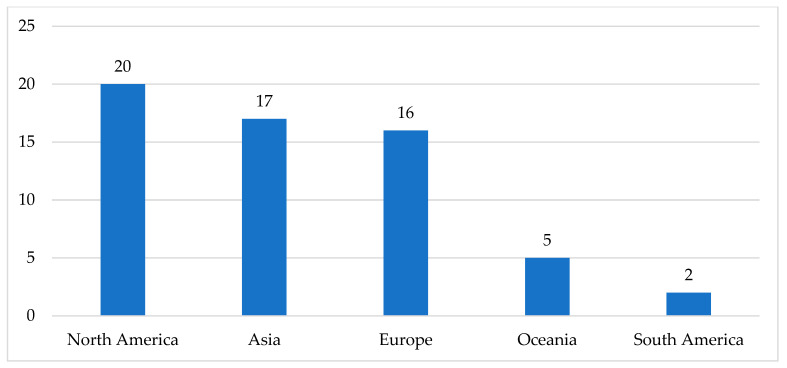
Areal distribution of studies (*n* = 60).

**Figure 5 healthcare-12-00957-f005:**
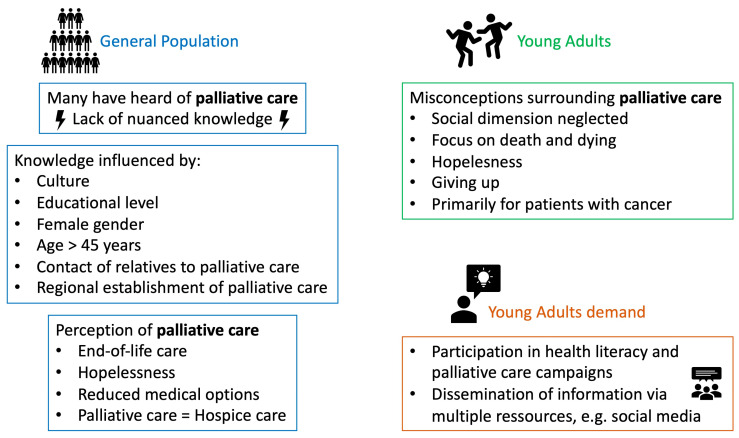
Graphical summary of the main findings.

**Table 1 healthcare-12-00957-t001:** PICOS process.

PICOS	Inclusion Criteria	Exclusion Criteria
P—Population	General population International No history of severe or life-limiting illnesses	Palliative patients Healthcare professionals
I—Intervention	If available (not necessary): Public relations campaigns Educational work Information dissemination	(not applicable)
C—Comparison	(not applicable)	(not applicable)
O—Outcomes	Perception of palliative care Knowledge of palliative care	Work that does not deal with the primary endpoints.
S—Study design	Randomized controlled trials Qualitative studies Review papers Systematic reviews Scoping reviews	Editorials Letters to the editor Statements

## Data Availability

The original contributions presented in the study are included in the article and Appendix A; further inquiries can be directed to the corresponding author.

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
