# Peer review of "The General Public and Young Adults’ Knowledge and Perception of Palliative Care: A Systematic Review"

_healthcare, 2024, doi:10.3390/healthcare12100957_

Round 1

Reviewer 1 Report

Comments and Suggestions for Authors

Short Note in General

The topic of the systematic review is extremely relevant and timely, especially in light of demographic changes, as the prevalence of people living with chronic and/or terminal illnesses is increasing. The findings can serve as a basis for developing relevant educational policies and programs. It is a paper with a fluid speech and a low rate of plagiarism.

Strengths

Timeliness and Relevance: The manuscript discusses a topic becoming increasingly significant in public health due to demographic shifts and the growing prevalence of chronic conditions. It aptly explains the necessity of palliative care in the context of rising healthcare demands.

Comprehensive Approach: The review is thorough and focused, employing multiple databases and including grey literature to compile the most complete dataset and analysis possible.

Theoretical and Practical Integration: The use of stigmatization models to provide theoretical explanations for the findings adds depth to the conversation by connecting data results to general sociological theories.

Areas for Improvement

Clarity and Focus: The manuscript could benefit from a clearer distinction of results for the general population versus young adults. While the research differentiates these groups in the analysis, a clearer opening could help improve the logical flow of ideas, especially for unspecialized readership.

Comparative Analysis: Adding more comparative elements between demographic groups or countries, considering the global scope of this literature review, could be useful.

Methodological Assessment: The inclusion and exclusion criteria are clear; however, the review could provide more details about the narrative synthesis process, including how discrepancies between reviewers were resolved.

Specific Suggestions

Expand on Interreligious Dialogue: You should discuss how different religious and cultural backgrounds influence perspectives on palliative care, informing more culturally relevant public health interventions. There are lots of relevant bibliography materials in this regard to be cited and used in this regard.

Potential Challenges: Consider potential difficulties in establishing educational programs on palliative care, especially in regions with developing health infrastructure.

Future Directions: Provide possible public health initiatives or educational programs that could raise awareness and knowledge of palliative care among young adults.

Conclusion

This systematic review is exceedingly relevant and vital from a public health perspective, as it underscores the gap between the need for palliative care and the public’s limited understanding of its necessity. By targeting young adults, this manuscript identifies a critical demographic that could help shift long-term perceptions of palliative care. I warmly recommend it for publication in your journal.

Comments on the Quality of English Language

The article is well-written with a couple of minor issues in clarity that could be easily addressed with slight restructuring [in some phrases' topic] - nothing to be worried about; editors can help.

Reviewer 2 Report

Comments and Suggestions for Authors Yann-Nicolas et al. assess the views and comprehension of palliative care among the general public and young adults (18–24). They highlight the stigma around palliative care and the need for de-stigmatization campaigns. The study's objective is to examine the current level of knowledge regarding palliative care in various age groups and geographical locations. It emphasizes the importance of removing barriers and expanding access to palliative care, especially for young adults. Taken as a whole, the included literature in this systematic review is current, contains solid, comprehensive information, and conforms with publication requirements. I do recommend a small modification, though. The writers should think over the following remarks. 1. Lines 112–114 state that the author described the screening procedures used by two separate researchers to evaluate the literature. But why wouldn't the results reported by the two researchers match if the screening criteria were maintained consistent? Could the outcomes alter if a single researcher conducted two such screenings independently? If everything is as it should be, then either the screening criteria are wrong, or the tools employed throughout the screening process may have had an impact on the outcomes. 2. The writers stressed on pages 107–108 that the German search string was created. Why include literature in German? How about we embrace French and Spanish? I believe that these languages were more widely used than German. Is this field's study led globally by Germany? Or is it only that the writers are German? 3. In Figure 2, there is no unit declaration for the x-axis. 4. Because images 2, 3, and 4 are bar figures and depict comparable contexts, I advise you to make them subfigures within a figure. 5. Since the results in sections 3.2 and 3.3 are only discussed in writing, the results there should be displayed graphically. A word description is inconvenient for readers to understand this area of the text completely and intuitively.  

Reviewer 3 Report

Comments and Suggestions for Authors

Journal of Healthcare

Systematic Review Article;

The article entitled The General Public and Young Adults’ Knowledge and Perception of Palliative Care: A Systematic Review’’. The authors tried their best to investigate demographic change, chronic and oncological diseases gain importance in the context of public health. Palliative care plays a crucial role in maintaining the quality of life of those affected. International guidelines demand access to palliative care not only for the elderly but also for younger people who face severe illnesses. The author targets assessing the understanding and viewpoints regarding palliative care among the general population and young adults. As the general public still lacks a detailed understanding of palliative care. Palliative care faces stigma at multiple levels, leading to barriers when aiming at implementing it. Addressing young adults as a crucial peer group can help break down barriers and promote access to palliative care.

Comments for Authors

Write the keywords in alphabetical order.

“Although all results showed that young people are not equally informed about palliative care as older population groups, this was generally not the specific main research question that was focused on in the literature”. As this statement is correct could the author also focus on the lifestyle and literacy rate of the areas or population? Because it is directly related to palliative care or other health care.

What is the novelty of the study?

The author needs to revise the manuscript, as there are many grammatical mistakes and spilling mistakes.

Reviewer 4 Report

Comments and Suggestions for Authors

This systematic review aimed to assess the current knowledge and perception of palliative care among the general population and young adults aged 18-24 years. The authors found that the general public still associates palliative care with death, dying, and end-of-life care, primarily for elderly cancer patients. Knowledge of palliative care varies based on factors such as education level, income, gender, personal experience with palliative care, and the duration of established palliative care structures in a country. Ethnic minorities tend to have less knowledge about palliative care compared to majority groups. Young adults, while often familiar with the term "palliative care," lack a nuanced understanding of its content and associate it with death, dying, and care for the elderly. However, young adults with higher educational aspirations tend to have more knowledge about palliative care. The authors argue that addressing young adults as a target group in public health campaigns can help break down barriers and promote access to palliative care, as their health behavior is still susceptible to change.  

Comments

1. The majority of the included studies were cross-sectional, providing only a snapshot of knowledge and perceptions at a single point in time. Longitudinal studies could help identify changes in attitudes and knowledge over time.

2. The studies included in the review were from various countries and cultures, which may have different healthcare systems and cultural attitudes towards palliative care. This heterogeneity could make it difficult to draw generalized conclusions.

3. The review found few studies that evaluated the effectiveness of interventions designed to improve knowledge and perceptions of palliative care. More research on the impact of educational campaigns and other interventions would be valuable.

4. The review may be subject to publication bias, as studies with positive results are more likely to be published than those with negative or null findings.

5. The authors did not report a formal quality assessment of the included studies, which could help identify potential biases and limitations in the primary research.

6. While the review acknowledged the importance of cultural factors in shaping attitudes towards palliative care, a more in-depth exploration of these factors and their implications for designing interventions would be beneficial.
